# *Crocus heuffelianus*—A New Species for the Bulgarian Flora from Series *Verni* (Iridaceae)

**DOI:** 10.3390/plants12132420

**Published:** 2023-06-22

**Authors:** Tsvetanka Raycheva, Kiril Stoyanov, Samir Naimov, Elena Apostolova-Kuzova

**Affiliations:** 1Department of Botany and Agrometeorology, Agricultural University, 4000 Plovdiv, Bulgaria; raicheva.tzveta@gmail.com; 2Department of Plant Physiology and Molecular Biology, University of Plovdiv, 4000 Plovdiv, Bulgaria; samir.naimov@gmail.com (S.N.);

**Keywords:** *Crocus heuffelianus*, Bulgarian flora, ITS1/2 region, leaf anatomy, morphology

## Abstract

In the Pirin Mountains, at an elevation of around 1000 m, three populations of a new species of Bulgarian flora from the genus *Crocus*, series *Verni*, were discovered. The species was compared to the morphologically related *C. veluchensis,* and presented with diagnostic morphological and anatomical features. Despite the high degree of morphological similarity, the molecular analysis, which included sequences from all related species (*C. cvijicii*, *C. dalmaticus*, *C. jablanicensis*, *C. rujanensis*, *C. sieberi* subsp. *atticus*, and *C. veluchensis*), distinguished the Pirin Mountains’ populations, and revealed the closest relationship to *C. heuffelianus*. Despite the *C. heuffelianus/C. verni* complex’s uncertain taxonomic status, our findings on the local population, based on morphometric, anatomical, molecular, and geographic analyses, indicate its belonging to the putative allotetraploid *C. heuffelianus* of south-eastern Europe and the Balkans, and an expansion of its range to the southeast. Given the taxonomic uncertainty and unclear phylogenetic relationships of the taxa in the *Crocus vernus* complex, we considered it appropriate to accept our taxon as *Crocus heuffelianus*. So far, only *C. tommasinianus* Herb. has been found in Bulgarian flora from the *Crocus* series *Verni*, but in terms of altitude and morphological features, the species from our collection is close to the Balkan endemic *C. veluchensis*, which belongs to the *C. sieberi* aggregate. Morphologically, it differs by the dark, heart-shaped spots on the tip of the tepals, and the presence of one bract. A detailed comparative anatomical analysis between the three species of crocuses from the series *Verni* in Bulgaria shows discrete differences: the width of the white stripe and lacunar area are good distinguishing features, as are the number of conducting vessels.

## 1. Introduction

The genus *Crocus* contains over 235 species worldwide [1]. One of the centres of biological diversity is the Balkan Peninsula [2]. In Mathew’s (1982) monograph, the genus was divided into the sections *Crocus* Mathew (series *Verni* Mathew, *Scardici* Mathew, *Crocus* Mathew) and *Nudiscapus* Mathew (series *Reticulati* Mathew, *Biflori* Mathew, *Flavi* Mathew) [2]. There are controversies and uncertainties related to the systematics and phylogenetic relationships within taxonomic groups. One critical group is the *Verni* series, with mostly spring-flowering crocuses from central and southern Europe, some of which are important ornamentals [3]. So far, *C. tommasinianus* Herb is the only known Bulgarian flora from the series *Verni*. In February 2023, a large population of a species unknown to our flora, with a V-shaped to heart-shaped spot on the tips of their purple tepals, was discovered above the town of Bansko. The species was determined to be the closest in comparative morphological characters to *Crocus heuffelianus* Herb., which belongs to the series *Verni*. During field surveys in the Pirin region, distant from the first population, two more populations of the same taxon were observed. In terms of altitude, ecological conditions, and morphological features, our species is close to the Balkan endemic *C. veluchensis* Herb., which belongs to the *C. sieberi* aggregate [4,5]. *Crocus vernus* (L.) Hill. is a European alpine species with a general distribution west to the Pyrenees and Alps, east to Ukraine, and south to Sicily and Albania [2,6]. Mathew accepted *Crocus heuffelianus* in the synonymy of *C. vernus* ssp. *Vernus*, and has also included *C. scepusiensis* (Rehm. et Wol.) Borb., *C. napolitanus* Mord. Laun et Lois, and *C. purpureus* Weston within it. [2]. However, more recent studies support the recognition of *C. heuffelianus* as a distinct taxon [7,8,9,10,11].

*Crocus heuffelianus* is distributed east to Austria, throughout the Balkan Peninsula, with the southern border in Albania [7]. It extends east into Romania and Ukraine and also extends into northeast Italy [8], with the northern distribution limit in Poland [6,9,10]. *Crocus heuffelianus* is closely related to several taxa with ambiguous taxonomic ranks accepted by different authors, e.g., *C. scepusiensis* (Rehmer & Wol.) Borbás ex Kulcz., *C. vittatus* Schloss. & Vuk., nom. illeg. *Crocus vernus* is a European alpine species, with a general distribution in the Pyrenees and Alps, east to Ukraine, and south to Sicily and Albania [2,6]. Recent phylogenetic analyses of polyploid lines from *C. heuffelianus* [3] testify to an allotetraploid origin resulting from the multiple reciprocal crosses between diploid genotypes of *C. heuffelianus* and *C. vernus*.

A good taxonomic marker in crocuses can be found in the literature at the level of leaf micromorphology [11,12,13,14,15,16,17,18]. Comparative anatomical studies of leaves in different taxa of the *Verni* series with different geographical origins have been carried out [18,19].

The phylogenetic relationships in the group are unclear. There are difficulties related to species differentiation, nomenclature, synonymy, and taxonomic affiliation [3,18]. A study of the populations of ser. *Verni* by morphology, anatomy, karyology, and molecular genetic markers proves allopolyploidy as a result of introgression between *C. heuffelianus* and *C. vernus*. The range of *C. heuffelianus* s.str. is in the Carpathians, and the western Balkans have occurred only tetraploids of *C.* cf. *heuffelianus* [3].

Due to taxonomic difficulties and unclear species status, a comparative morphological and anatomical study of the newly discovered taxon was made with the morphologically similar species *C. veluchensis* and the closely related *C. tommasinianus*. On the other hand, morphological differences do not always reflect kinship relationships. For this reason, we undertook sequencing of the internal transcribed spacer region (ITS: ITS1 + 5.8S rDNA + ITS2), which has already been successfully used in phylogenetic studies in the genus *Crocus* [20,21].

## 2. Results

### 2.1. Morphological Description Based on Bulgarian Materials

#### *Crocus heuffelianus* Herb., J. Hort. Soc. 2: 273 (1847) [22]

*Crocus heuffellianus* is a spring synanthous geophyte (Figure 1). The plant is 12–25(–32) cm tall. The corm is ovate-rotundate, with a diameter of 1.2–2.4 cm; with tunics of parallel fibres, unclearly reticulated on the tip; and without separating basal rings. Prophylls are missing. Kataphylls are present, 2–3(–4), white. The leaves are 2–3, in bloom shorter than the flowers, prolonging later, 2.8–4.4 mm wide. Bracts are present. Bracteoles are not present. The flowers are 1–2. The colour of the perigone is violet to purple, rarely pale, as the perigone segments (tepals) have a V-shaped or heart-shaped spot on the tip. The throat is visibly pubescent inside. The filaments are white. The anthers are yellow with white connectives. The stigma is exerted above the anthers, reddish orange, divided into three spathulate with wide lobes, each one indistinctly toothed. The capsule is ellipsoid, 12–14 mm long, 7–10 mm wide. The seeds are rounded, reddish, 1.8–2.6 mm in diameter, with distinct margins and a slightly developed caruncle (Figure 2A).

The key parameters, compared with those of *C. tommasinianus* and *C. veluchensis*, are shown in Table 1. The species is morphologically similar to *C. vernus*. The key parameters for distinguishing *C. heuffelianus* from *C. vernus* are compared with the Bulgarian populations in Table 2. 

### 2.2. Distribution and Habitats

Three localities of *C. heuffelianus* were observed, all in the Pirin Mountains (north) floristic region (Figure 3). The populations share the same distribution as *C. veluchensis*, but they remain intact. Mixed populations of these two species were not found. The populations were found in temporary or often humid meadows and open places, or on the borders of beach and pine forests, where there is a seasonal dominant of the early spring vegetation with a projective covering of between 3 and 5% (Figure 4). The first locality was found on the urbanized margin of the town of Bansko. The habitat was open land, a former forest of *Pinus sylvestris.* The other two localities were found on wet meadows in mixed forests of *Fagus sylvatica* and *Pinus sylvestris.* The registered accompanying species were: *Crocus chrysantus* Herbert (Herbert), *Scila bifolia* L., *Ficaria verna* Huds., *Alchemilla* sp., *Achillea* sp., *Erythronium dens-canis*, *Viola* sp., *Sanicula europaea* L., *Corydalis* sp., *Oxalis acetosella* L., *Gagea reticulata* (Pall.) Schult. & Schult. f. mosses, and grasses of Poaceae in an early phase.

The localities where *C. heuffelianus* occurred were outside the territory of the Pirin National Park, on unprotected land. The population in Bansko was on the territory of the town, on one of the former meadows still without building activities. The other two localities were on the bank of the Desilitsa River, under slight anthropogenic pressure (namely roads and extensive tourism). 

### 2.3. Leaf Anatomy

The anatomical features of *C. heuffelianus* from the evaluated populations are presented in Table 3.

The cross-sections of the leaves of *C. heuffelianus, C. tommasinianus*, and *C. veluchensis* showed distinguishable bifacial profiles (Figure 5). The typical white stripe in *Crocus* (lacuna area) has a different ratio to leaf width. The ratio between the white stripe and the section width is 1/5 in *C. tommasinianus* and 1/4 in *C. heuffelianus* and *C. veluchensis*. 

The keel of *C. heuffelianus* is almost rectangular. The widest vascular bundles are concentrated on the keel base. The keel base has almost straight angles, without ribs (Figure 6). This makes the white stripe wider than the keel base.

The parenchyma of the shoulders is uniform in the three species, with a slight difference in the rows of cells in the palisade area. Most of the sections of *C. tommasinianus* show one row of palisade parenchyma, rarely with a second layer with shorter cells. The palisade parenchyma of both *C. heuffelianus* and *C. veluhensis* has two layers of cells, rarely only one. The spongy parenchyma does not show any differences.

The count of the vascular bundles varies between 21 and 27, and does not show notable differences. The difference is in the ratio between the large vascular bundles (Figure 7). The vascular bundles are collateral, situated on one layer. The larger bundles of *C. heuffelianus* are four: two at the edges (terminal) and two at the keel (the largest). *Crocus veluchensis* has six larger vascular bundles: two terminal bundles, two on the bottom of the keel, and two at the blade near the lacuna area (the corner bundles). Similar is the ratio of the vascular bundles in *C. tommasinianus*, but not as clearly visible, and sometimes with larger bundles in the blade. According to the broadest bundles in the cross-section, *C. veluchensis* has the largest terminal vascular bundles, and this size is due to the larger cap of sclerenchyma. *Crocus heuffelianus* has the smallest terminal vascular bundles, according to the maximum size in the cross-section. The corner vascular bundle is 20% of the maximum in *C. heuffelianus*, as the other species have a higher value of this characteristic (Table 3, Figure 8).

The stomata are of the anomocyte type [13], arranged on the abaxial surface of the leaf blade and both lateral sides of the keel. The three species have sporadic single stomata on the adaxial epidermis. The three examined species have similar stomatal indexes. The major difference is in the shape of the cells of the abaxial epidermis (Figure 9). Those of *C. tommasinianus* have wavy borders, while the other species have straight cell borders.

### 2.4. ITS1/2 Region and Phylogenetic Tree

The alignment of the ITS1/2 regions of the evaluated species showed significant differences between *C. heuffelianus* and *C. veluschensis*, and similarities to the samples of both taxa found in the NCBI Nucleotide database (Table A1). The difference between the collected samples is visible, e.g., the big gaps in ITS1/2 in the group of *C. heuffelianus* (Figure A1: positions 33, 37, 397, 481, 503, and 530), as are the other evaluated samples, following the accepted taxonomy. The difference between *C. heuffelianus* and *C. tommasinianus* is visible as point replacements (Figure A1): the transitions G/A (positions 33, 37) and T/C (position 116), and the transversion T/A (position 439).

The phylogenetic tree (Figure 10) follows the phylogeny published before [20,21]. The samples of ser. *Verni* are grouped in their clade, as are the samples of *C. veluchensis* collected in the neighbourhood with *C. heuffelianus* (SOA 063356, Figure 10, OQ920186), which are grouped in another one. While the branch of *C.* agg. *sieberii* contains unconstrained samples of *C. veluchensis*, our samples of this species remain grouped in a subclade, which also contains the sequence of a sample from Hungary (HE801155). The rest of the sequences are grouped in the clade of *C. sieberii*, which also contains *C. rujanensis*, *C. atticus, C. dalmaticus, *and* C. robertianus.*

## 3. Discussion

The morphological features of the Bulgarian specimens coincide with the morphological data for *C. heuffelianus* [1], except for the pubescence mark on the perigonal tube. By this parameter, our samples match the Balkan populations of *C.* cf. *heuffelianus* [19].

The corm tunics of the three species are fibrous, without detaching rings, and with unclear interweaving at the tips. The heart-shaped purple spot on the tepal tips is a key morphological feature of *C. heuffelianus*. The quantitative differences between the tepal sizes are not discrete, but the largest tepals are those of *C. heuffelianus* (33–58 × 7–8 mm), followed by *C. veluchensis* (18–55 × 6–7 mm), and *C. tommasinianus* (22–43 × 6–7 mm).

A significant feature is the lack of bracteoles and the presence of a bract in* Verni*, as well as the fact that *C. veluchensis* has bracteoles but no bracts. The tinted tepal tip is a constant feature in the observed populations. Despite the data for a smooth perigone throat in *C. heuffelianus* [6,10], the collected plants from Bulgaria have pubescent throats, supporting the idea that the Balkan populations of *C. heuffelianus* have pubescent throats [20]. The shape of the stylodia is another significant difference between *C. heuffelianus* and *C. tommasinianus* (Figure 11).

The size of the fruit capsules in the evaluated species is too variable (Table 1). Therefore, they do not have a high diagnostic value.

Although seed morphology is highlighted as a diagnostic feature in *Crocus* taxonomy [23], information is available in the literature [24] for a small number of taxa. The difficulties are connected with the fact that the ripening of the seeds is between 3 and 5 months after the bloom [25], and that is why the collection in natural populations is difficult. Therefore, we have carefully obtained them from planted specimens. The seed macromorphology, sculpture, and ornamentation in *C. heuffelianus*, *C. tommasinianus*, and *C. veluchensis* (Figure 2) show clear differentiation by groups. The caruncle of *C. veluchensis* is over 1 mm, while those of *C. heuffelianus* and *C. tommasinianus* are visibly smaller, up to 0.5 mm. This corresponds with the position of *C. heuffelianus* in the section *Verni* together with *C. tommasinianus*, and the outstanding position of *C. veluchensis*. 

The anatomical diagnostic features of *C. veluchensis* are the thicker spongy parenchyma, narrower white stripe, lack of hairs on the keel, and bigger terminal vascular bundles. These parameters, together with the height of the keel, are bigger in the leaves of *C. heuffelianus, *as noted in the populations from Serbia [18]. The epidermis of *C. tommasinianus* stands out with clearly undulating tangential walls, while the cell walls of *C. heuffelianus* and *C. veluchensis* are straight or slightly undulating (Figure 9). The count of the stomata on the abaxial epidermis variates between 200 and 400 per mm^2^. Their sizes are similar—about 20 × 31 μm, making these characteristics unreliable for taxonomic purposes.

A commonly used characteristic is the width of the white stripe running axillary along the central part of the leaf as a ratio to the diameter of the leaf [25]. This ratio in the samples of *C. tommasinianus* is 1/5, while in *C. veluchensis* and *C. heuffelianus *it is 1/4. More exact differences could be seen in the ratio between the white stripe and the base of the keel. The stripe of *C. heuffelianus* is usually wider than the keel, as the white stripes of *C. veluchensis* and *C. tommasinianus *take up 80% (4/5) of the width of the keel. 

Another important difference is the shape of the keel (Figure 6). The keel of *C. heuffelianus* is almost rectangular. In comparison, the keel of *C. tommasinianus* has two thin ribs on the bottom. The two species above have keels with a straight base. In contrast, the keel of *C. veluchensis* is widely arcuate, with a central vascular bundle placed in the curve. The keel angles of *C. heuffelianus* and* C. tommasinianus* are covered with papillae, which can be seen on the whole surface of the leaf, just as the keel angles of *C. veluchensis *look without papillae.

The total number of vascular bundles in a cross-section could be a significant taxonomic feature [14]. The number of vascular bundles in *C. heuffelianus* varies more (19–42) than in *C. tommasinianus* (20–25) and *C. veluchensis* (21–28). Despite the wider variation in *C. heuffelianus, *this parameter could not be used as a single criterion. Combined with ratios of the major vascular bundles (terminal, keel, and corner towards the larger size), it could set a usable way to recognize *C. heuffelianus* (average 49%, 78%, 16%), *C. tommasinianus* (71%, 80%, 39%), and *C. veluchensis* (64%, 37%, 37%). This fact demonstrates the close relationship between both taxa. According to the literature [1], these species are very different in morphology and karyology. The collected samples from Bulgaria are in agreement with the description of *C. heuffelianus* (Table 2). An exception is the indumentum of the throat, as noticed in all Balkan populations of this species [18,19]. 

The major features in the ITS1/2 regions of the evaluated species are the types of conservative nucleotide differences. The species Ser. *Verni* can be recognized by only three mutation points in the evaluated part of the ITS1/2 region: two constant transitions and one transversion. The ITS sequences of the collected samples are identical to the ITS sequence of a specimen with origins in the Western Balkans. Additionally, the aligned sequence regions of all samples of *C. heuffelianus* are completely identical to those of *C. vernus*, as noticed before [3]. 

Currently, there is no information regarding the chromosome number of the Bulgarian populations of *C. heuffelianus*. According to the morphology, leaf profile, and ITS similarity with *C. vernus*, the observed specimens are similar to those described as tetraploid *C.* cf. *heuffelianus* from the Western Balkans [3], and suggest an expansion of the populations of this taxon to the southeast could be inferred.

The evolutional events (substitutions, insertions, and deletions) between Ser. *Verni* and our samples of *C. veluchensis* are illustrated by big gaps in the sequences. The lack of such evolutionary events inside ser. Verni illustrates the close relationship between *C. heuffelianus, C. vernus,* and C. *tommasinianus.* They are young taxa with a common ancestor, and *C. veluchensis* is not their sister taxon.

The phylogenetic tree (Figure 10) confirms the phylogeny published before [20,21]. It also proves the presence of *C. heuffelianus *sensu lato in the flora of Bulgaria. Following the reason above, the members of Ser. *Verni* are grouped in a single branch, as are the samples of *C. veluchensis*. All samples of *C. veluchensis* from Bulgaria have almost identical ITS1/2 sequences with that of a referent specimen from Hungary. The other samples, annotated as *C. veluchensis,* stand in another branch. 

The evaluated taxon is new to the country, and is morphologically close to the description of *C. heuffelianus* sensu lato. The species have been neglected because of the visual similarity with *C. veluchensis*, and the insufficient investigations in the genus *Crocus* in Bulgaria.

Given the presence of polyploidy in ser. *Verni*, further studies of Bulgarian representatives should be focused on karyological aspects, namely chromosome number and genome size.

## 4. Materials and Methods

### 4.1. Evaluated Specimens

The study was based on fresh individuals collected from Bulgaria: three populations of *C. heuffelianis*, one of *C. tommasinianus,* and eight of *C. veluchensis*. *Crocus tommasinianus* is protected by Bulgarian environmental law. In this case, the plants were collected after permission from the Bulgarian Ministry of Environment and Water (authorization 871/24.04.2021). For comparative specimens, plant materials were used from the herbaria SOA (Agricultural University, Plovdiv) and SOM (Institute of Biodiversity and Ecosystem Research, Bulgarian Academy of Science), as well as high-resolution images from foreign herbaria downloaded from GBIF.org [26,27,28,29,30,31,32]. The distribution map of Bulgarian specimens (Figure A1) was built using QGIS [33] on a layer from http://openstreetmap.org (accessed on 8 November 2022).

The morphological analysis was made on 30 fresh individuals of *C. heuffelianus*, compared with fresh and dry specimens of *C. tommasinianus* and *C. veluchensis*. The metric values were measured using a calliper. The images of seeds and small parts were photographed using a stereomicroscope (Leica EZ4W). 

The fresh specimens collected in this study are represented below by floristic region (subregion in brackets), MGRS square, locality description, decimal coordinates (if available), altitude, date (collectors), herbarium acronym [34], and specimen numbers. All evaluated specimens are described in Appendix B. The compared specimens from images are cited with their origin databases.


*Crocus heuffelianus*


BULGARIA: Pirin (North): 34TGM03: Bansko, 1020 m, N41.821668 E23.476085, 2023-03-06 (G. Gerdjikov) SOA 063340; 1018 m, N41.82177 E23.47616, 2023-03-16 (coll. T.Raycheva and K.Stoyanov) SOA 063342 (GenBank OQ918262); 34TGM13: Between Dobrinishte and the hut Goce Delchev, on the bank of Desilitsa River, 1000 m, N41.796684 E23.567387, 2023-03-17 (coll. Ts. Raycheva and K. Stoyanov) SOA 063351 (GenBank OQ918261); 34TGM12: Between Dobrinishte and the hut Goce Delchev, under Mogilata Peak, 1015 m, N41.78486 E23.55544, 2023-03-17 (coll. T.Raycheva and K.Stoyanov) SOA 063348 (GenBank OQ918260).


*Crocus veluchensis*


BULGARIA: Stara Planina (Central): 35TLH03: Beklemeto Narrow, 1440 m, N42.767 E24.611, 2022-04-12 (coll. T. Raycheva and K. Stoyanov) SOA 063262;35TLH04: Beklemeto Narrow, 1270 m, N42.79673 E24.62651; 2019-04-04 (coll. N. Trifonov) SOA 062631; 35TLH53: Ispolin Peak, 1523 m, N42.7386111 E25.2522222, 2018-04-05 (coll. Y. Marinov) SOA 062486; Vitosha region: 34TFN81: above Akademika sport base, under Skoparnika Peak, 1905 m, N42.54838 E23.31165, 2018-04-21 (coll. K. Stoyanov) SOA 062475; under Trite Komini peak, 2223 m, N42.55867 E23.28389, 2019-06-10 (coll. K. Stoyanov) SOA 063121; 34TFN91: above the panoramic path, 1080 m, N42.60176 E23.3234, 2019-04-11 (coll. K. Stoyanov) SOA 062632; Pirin (North): 34TFM92: 34TFM93: Betolovoto locality, on the bank of Byala Reka, 1112 m, N41.84798 E23.39011, 2021-06-26 (coll. T. Raycheva and K. Stoyanov) SOA 063230, 2022-03-29 (coll. Ts. Raycheva and K. Stoyanov) SOA 063259; 34TFM94: Predela Narrow, 1054 m, N41.892357 E23.330803, 2023-03-17 (coll. T. Raycheva and K. Stoyanov) SOA 063350; 34TGM12: Between Dobrinishte and the hut Goce Delchev, under Mogilata Peak, 1015 m, N41.781837 E23.549747, 2023-03-17 (coll. T.Raycheva and K. Stoyanov) SOA 063356 (GenBank OQ920186); Rhodopi Mts. (West): 34TGM35: Artificial pine forest between Sveta Petka and Yundola, 1327 m, N42.046604 E23.869862, 2022-03-27 (coll. T. Raycheva and K. Stoyanov) SOA 063260; Open places over Avramovo Railway Station, 1270 m, N42.03593 E23.81844, 2023-03-16 (coll. T. Raycheva and K. Stoyanov) SOA 063341, 063343; 34TGM36. Meadows around Yundola, 1091 m, N42.0638889 E23.8475, 2022-04-22 (coll. T.Raycheva and K. Stoyanov) SOA 063228 (GenBank OQ920187); 35TKG84: Wet meadow between Ravnogor and Atolouka localities, 1354 m, N41.9578778 E24.3486111, 2020-06-19 (coll. T. Raycheva and K. Stoyanov) SOA 063208; Rhodopi Mts. (Central): 35TKG94: Around Vurkhovrukh Hut, 1980-04-26 (coll. M. Popova) SOA 041385; 1982-04-17 (coll. M. Popova) SOA 039223, 039833; 1488 m, N41.9611667 E24.5338889, 2019-03-24 (coll. T. Raycheva) SOA 062589 (GenBank OQ920183); 35TKG95: Between Peroushtitsa and Skobelevo, 960 m, N42.0162778 E24.5505833, 2019-03-24 (coll. T. Raycheva) SOA 062584 (GenBank OQ920184); 812 m, N42.02534 E24.54629, 2019-03-24 (coll. T. Raycheva) SOA 062588; 2020-04-19 (coll. T. Raycheva) SOA 062784; 35TLG04: the road to Stoudenets, N41.96896 E24.68819, 1420 m, 2019-04-11 (coll. G. Karaycheva and T. Raycheva) SOA 062634, 062635; 35TLG05: Boykovo, in the village, N42.00169 E24.62325, 1086 m, 2018-04-01 (coll. K. Stoyanov) SOA 062482; Around the Zdravets hut, 1185 m, N42.0041667 E24.6930556, 2018-03-10 (coll. Ts. Raycheva) SOA 062485 (GenBank OQ920185); 35TLG11: Rozhen narrow, 1425 m, N41.67148 E24.72673, 2021-06-11 (coll. T. Raycheva and K. Stoyanov) SOA 063158; Momchilovtsi 2022-04-06 (coll. V. Trifonov) SOA 063257.


*Crocus tommasinianus*


BULGARIA: Forebalkan (West): 34TFP14: Darkov-Dol locality, 378 m, N43.78833 E22.41325, 2020-03-13 (coll. T. Raycheva, K. Stoyanov and K. Uzundzhalieva) SOA 062847; 34TFP34: Yonov-Bair hill, 348 m, N43.76794 E22.72396, 2020-03-13 (coll. T. Raycheva, K. Stoyanov and K. Uzundzhalieva) SOA 062849; 34TFP15: Vagleshtarnika locality, 377 m, N43.80896 E22.39386, 2020-02-29 (coll. T. Raycheva, K. Stoyanov and K. Uzundzhalieva) SOA 062848.

### 4.2. Anatomical Investigations

Quantitative measurements and observations of qualitative features of fresh samples for the morphological analyses were conducted. To examine the leaf anatomy, 10 individuals per population, from 3 populations in total, were collected during the flowering time and conserved in 75% ethanol. Transverse cross-sections and epidermal areas of leaves from each individual were manually constructed from the middle part. The epidermis was peeled using a surgical scalpel. The microscopic slides were fixed with glycerine. The snapshots of the microscope slides were taken using the Leica 750 and Motic Panthera digital microscopes, while the measurements of the characters were completed using Micam 2.4 software [35]. The 24 anatomical features included section width and height, arm length, white stripe width, the number and size of vascular bundles, palisade tissue height, the height and width of palisade and spongy cells, and the height and width of the adaxial and abaxial epidermal cells. Abaxial epidermal cells, including the size of the stomata, were measured in the area of the stomatal rows (between the ribs). The values were expressed through the basic statistical parameters: mean, standard deviation, maximum, and minimum. 

### 4.3. Molecular Methods

The plant genomic DNA was purified as described using the DNeasy Plant Mini Kit (QIAGEN, Hilden, Germany), as described earlier [36]. 

The quality of the resulting DNA was assessed spectrophotometrically, using the Epoch™ Microplate Spectrophotometer, Agilent Technologies, Santa Clara, CA, USA, and DNA integrity was evaluated by 1% agarose gel electrophoresis.

The DNA fragment encoding for the ITS1–5.8S rDNA–ITS2 cluster was amplified using the following primers: ITS-A (5′–GGAAGGA-GAAGTCGTAACAAGG–3′) and ITS-B (5′–CTTTTCCTCCGCTTATTGATATG–3′) [37,38]. The reactions, set in a final volume of 50 μL contained 1x reaction buffer, 200 μM of dNTPs, 0.2 μM of each primer, 100 ng of genomic DNA, and one unit of Q5 High Fidelity DNA polymerase (New England Biolabs). The PCR amplification was conducted under the following parameters: initial denaturation at 94 °C for 45 s, followed by 30 cycles at 94 °C for 10 s for denaturation, 10 s at 62 °C for primer annealing, 30 s at 72 °C for primer extension, and a final elongation step of 2 min at 72 °C. Amplified PCR products were separated by 0.8% agarose gel electrophoresis, excised from the gel, and purified using a QIAquick Gel Extraction Kit (QIAGEN). The purified DNA fragments were bidirectionally sequenced in a Eurofins facility. All evaluated ITS sequences from *Crocus heuffelianus*, *C. veluchensis* and other referent species are described in Appendix A.

### 4.4. Phylogenetic Analysis

The obtained nucleotide sequences were blasted against those from the NCBI Nucleotide database [39,40]. The best hits were downloaded and used for phylogenetic analysis. The alignment of the sequences was achieved using ClustalW multiple alignments [41]. The result of the alignment was visualized in Figure A1 using the CLC Sequence Viewer [42]. The phylogenetic analysis was conducted using Bayesian phylogenetic inference with Mr Bayes 3.2 [43]. The parameters of the analysis were the same as described by Harpke et al. [21]: 2 × 4 chains for two million generations, nuclear data set ΓTP + G + I, sampling tree per 1000 generations, two independent runs. The result was visualized as a tree using TreeGraph 2 [44]. The analysis included 31 nucleotide sequences, cited as numbers of GenBank entries in the phylogenetic tree (Figure 10, Table A1). We accepted the series *Crocus* for the outgroup. 

## Figures and Tables

**Figure 1 plants-12-02420-f001:**
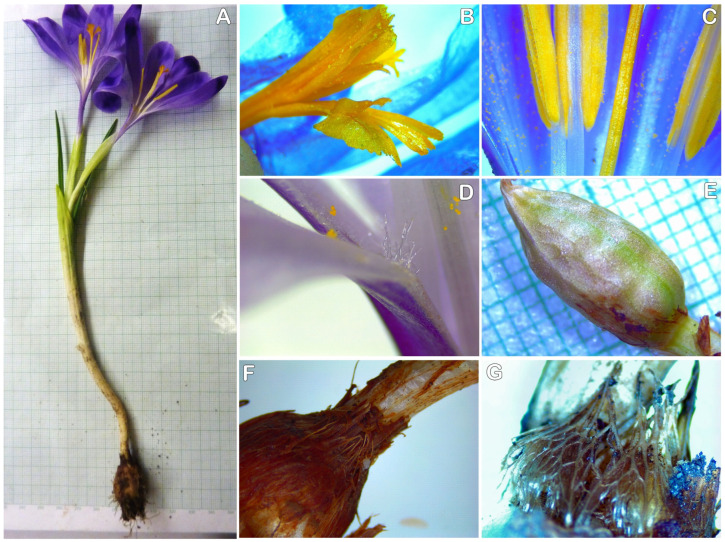
Morphology of *Crocus heuffelianus* (SOA 063351, grid 1 mm): (**A**) whole plant; (**B**) stylodia; (**C**) anthers; (**D**) hairs of the throat; (**E**) capsule; (**F**) corm neck; (**G**) tunics.

**Figure 2 plants-12-02420-f002:**
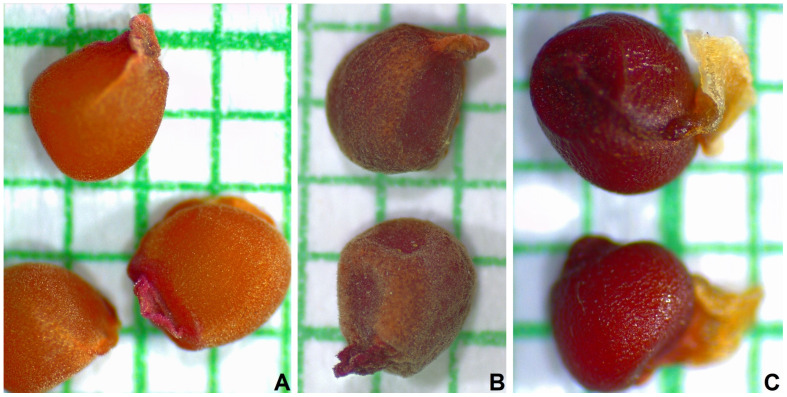
Seeds of (**A**) *Crocus heuffelianus*, (**B**) *C. tommasinianus* and (**C**) *C. veluchensis* (grid 1 mm).

**Figure 3 plants-12-02420-f003:**
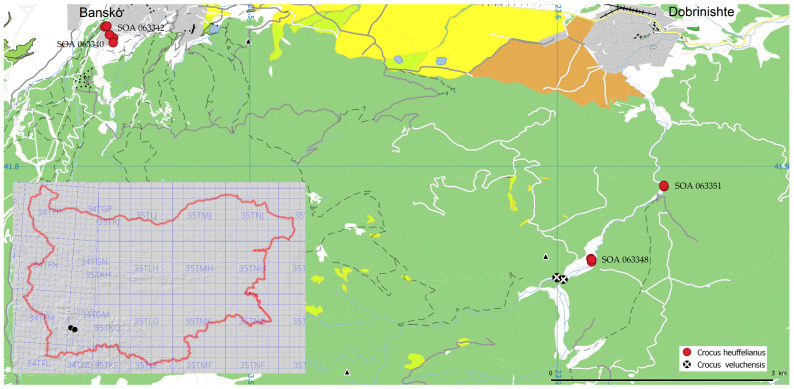
Observations of *Crocus heuffelianus* and *C. veluchensis* and position of the localities of *C. heuffelianus* in Bulgaria (the red dot numbers are explained in the text).

**Figure 4 plants-12-02420-f004:**
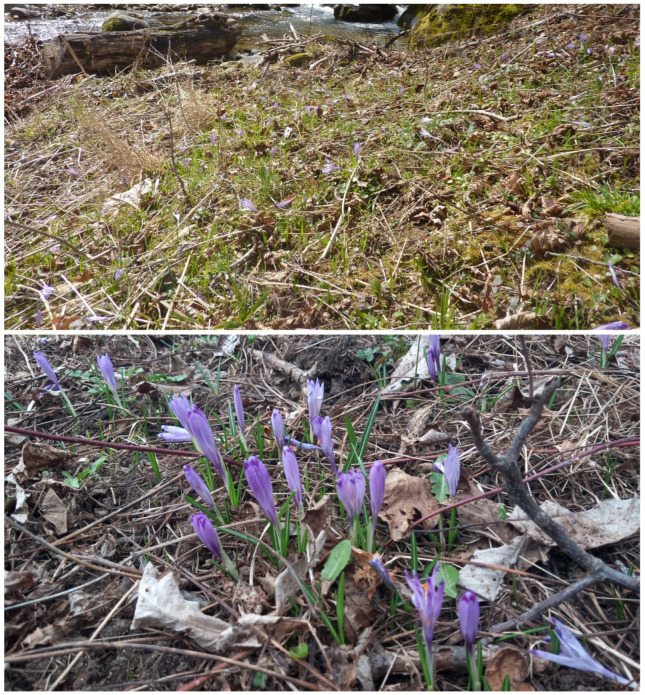
Habitat and group of *Crocus heuffelianus* (SOA 063351).

**Figure 5 plants-12-02420-f005:**
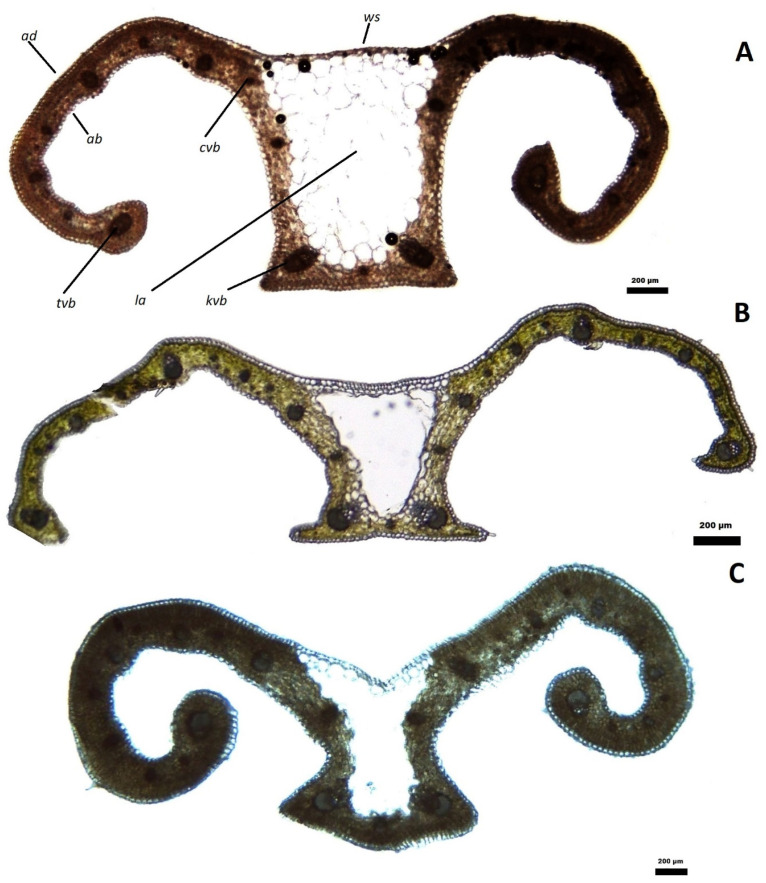
Leaf sections of (**A**) *Crocus heuffelianus*, (**B**) *C. tommasinianus* and (**C**) *C. veluchensis* (objective 4x, scale 200 μm): *ad*—adaxial epidermis, *ab*—abaxial epidermis, *la*—lacuna area, *ws*—white stripe, *tvb*—terminal vascular bundles, *kvb*—keel vascular bundles, *cvb*—corner vascular bundles.

**Figure 6 plants-12-02420-f006:**
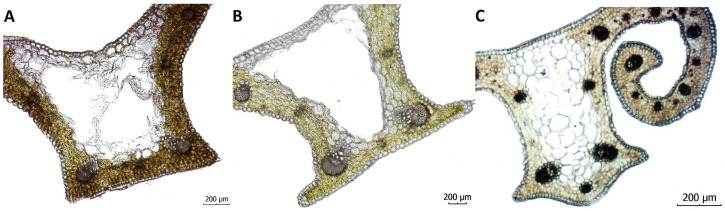
Keel shape of (**A**) *Crocus heuffelianus*, (**B**) *C. tommasinianus* and (**C**) *C. veluchensis* (magnification 10×, scale 200 μm).

**Figure 7 plants-12-02420-f007:**
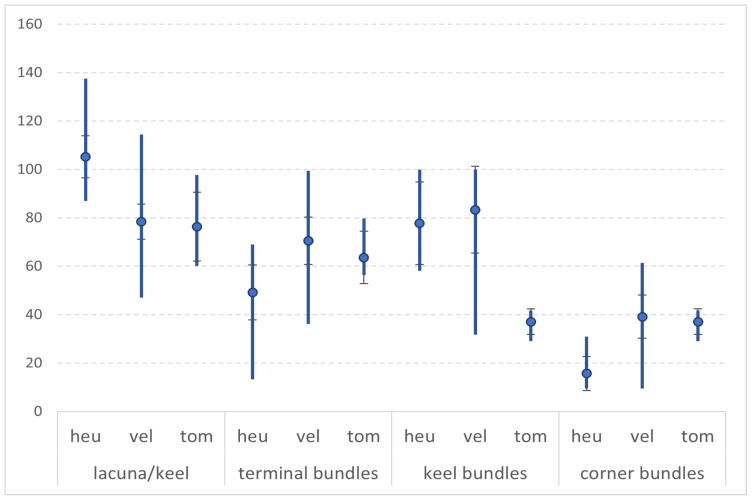
Ratios between the commented parameters (minimum, maximum, average, and standard deviation) in *C. heuffelianus* (heu), *C. veluchensis* (vel) and *C. tommasinianus* (tom).

**Figure 8 plants-12-02420-f008:**
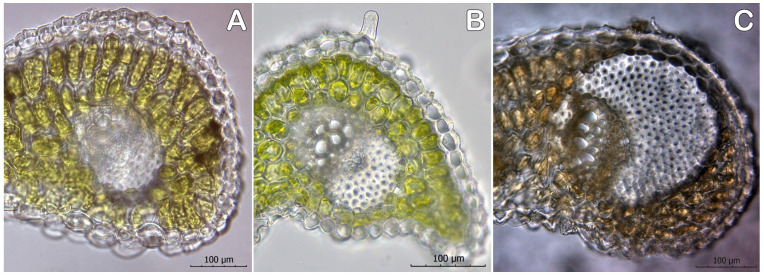
Terminal vascular bundle in (**A**) *Crocus heuffelianus*, (**B**) *C. tommasinianus*, and (**C**) *C. veluchensis* (magnification 40×, scale 100 μm).

**Figure 9 plants-12-02420-f009:**
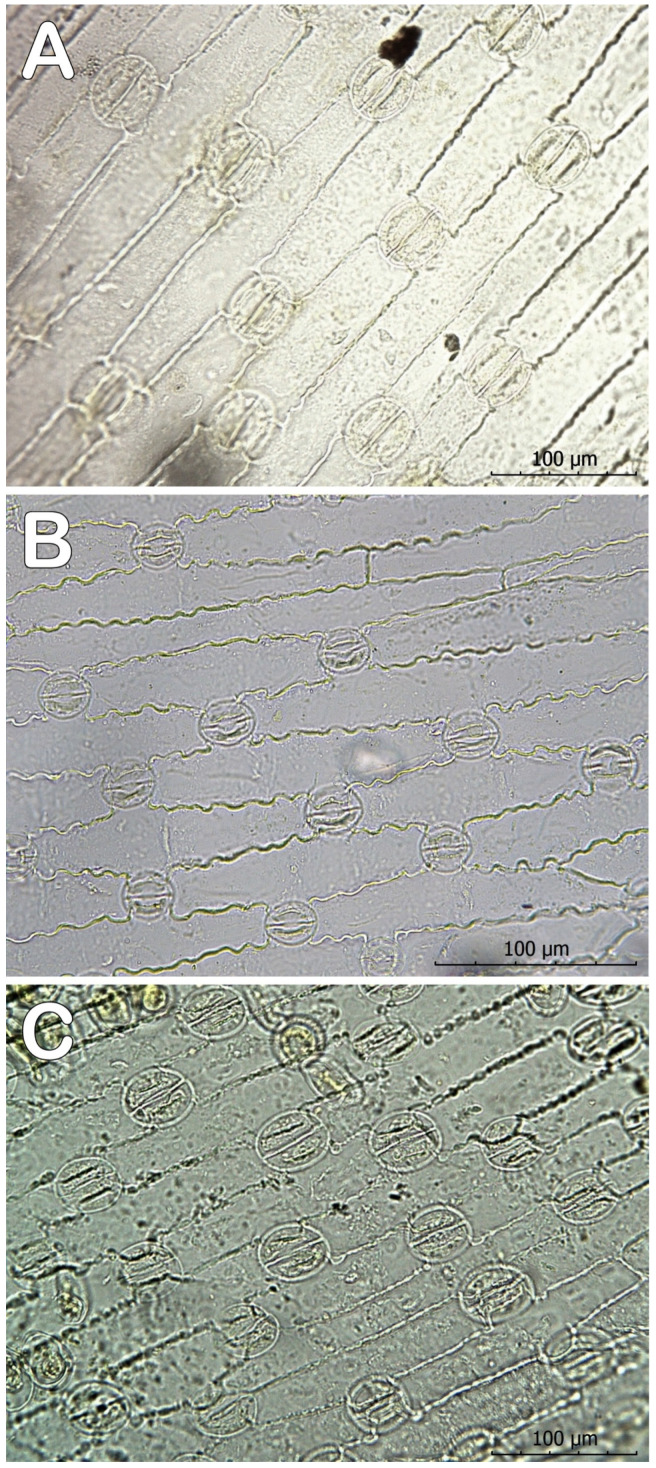
Abaxial epidermis of (**A**) *Crocus heuffelianus*, (**B**) *C. tommasinianus*, and (**C**) *C. veluchensis* (magnification 40×, scale 100 μm).

**Figure 10 plants-12-02420-f010:**
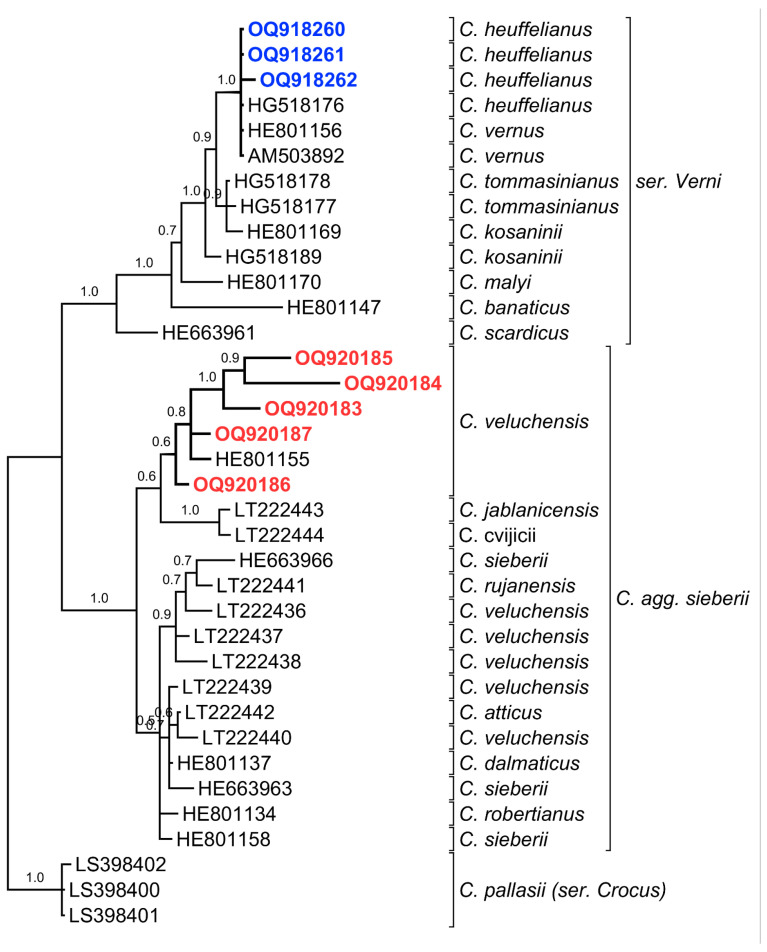
Placement of *Crocus heuffelianus* in the phylogenetic tree, obtained from a Bayesian phylogenetic inference of the nuclear rDNA ITS regions using the methodology of Harpke et al. [20]. Posterior probabilities are designated by numbers. See Table A1 for details.

**Figure 11 plants-12-02420-f011:**
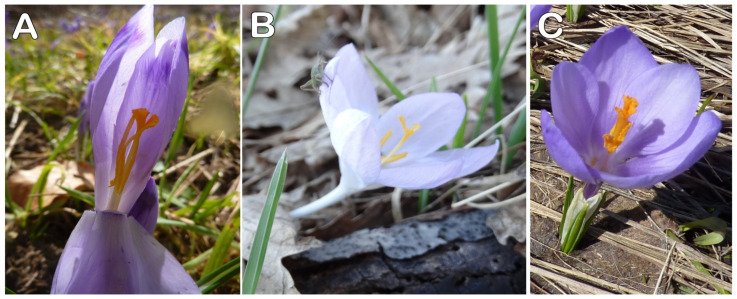
Flowers of (**A**) *C. heuffelianus*, (**B**) *C. tommasinianus*, and (**C**) *C. veluchensis*.

**Table 1 plants-12-02420-t001:** Morphological comparisons between the Bulgarian populations of *C. heuffelianus*, *C. tommasinianus* and *C. veluchensis*.

Characters	*C. heuffelianus*	*C. tommasinianus*	*C. veluchensis*
Corm	rounded, 12–24 mm	rounded 10–15 mm	rounded 10–20 mm
Tunics	parallel fibres	parallel fibres	parallel fibres
Bracts	present	present	missing
Number of bracteoles	missing	missing	present
Number of leaves	2 (3)	2 (3)	3–4 (5)
Flower colour	lavender purple, rarely pale, with a dark violet heart-shaped spot on the tip	pale purple	light to lavender blue, or dark purple
Perigone segments length/width	33–58 × 9–25 mm	22–43 × 12–18 mm	18–55 × 5–20 mm
Perigone throat	lilac to deep violet, pubescent	pubescent	pubescent
Filaments (mm)	10–14 mm, white	6–8 mm, white	8–12 mm, white
Anthers (cm)	18–22 mm	10–14 mm	12–18 mm
Style colour	yellow to orange	pale orange	yellow to orange
Style branched	3 spade-shaped lobes, exerted above the anthers	3 thin lobes, under the level of the anthers	3 spade-shaped lobes, exerted above the anthers
Capsule (mm)	12–14 × 7–10 mm	10–14 × 8–9 mm	10–12 × 5–7 mm
Seed (mm)	1.8–2.5 mm, slightly developed caruncle	2.2–4 mm, slightly developed caruncle	2.4–2.8 mm, big caruncle
Seed colour	reddish brown	reddish brown	dark reddish to brown

**Table 2 plants-12-02420-t002:** Morphological comparisons of the Bulgarian populations of *C. heuffelianus* with the morphological descriptions of *C. heuffelianus* and *C. vernus* from the literature [1].

Characters	*C. vernus* (According to Rukšāns [1])	*C. heuffelianus* (Bulgarian Specimens)	*C. heuffelianus* (According to Rukšāns [1])
Tunic neck	Up to 10 mm long, occasionally longer, somewhat bristly, formed by irregular prolonged fibres of the main tunic	Short, 2–5 mm, formed by a bunch of tunic fibres	Short, 2–3 mm long, formed by a bunch of tunic fibres
Perigone throat	Pubescent or hairy, mostly white, sometimes slightly shaded greyish or light violet	Lilac to deep violet, pubescent	White, nude
Perigone segments size	24–32 mm × 9–12 mm	33–58 × 9–25 mm	(23–)31–43(–55) mm × (7–)11–16(–22) mm
Outer segments	Plain white or blue, or with a darker stripe, without a V-shaped blotch	Darker than the inner, with a distinctly V-shaped blotch	Darker than the inner, with a distinctly V-shaped blotch
Inner segments	Sometimes more striped	Lighter than the outer,	Lighter than the outer, more often emarginate
Filaments length	(5-)7–10 mm	10–14 mm	12–14 mm
Anthers	(5–)8–12 mm, as usual same length as the filaments	10–14 mm	12–16 mm, a little longer than the filaments
Style colour	Orange to red	Yellow to orange	Yellow to orange
Style length	Ending deep within the anthers, rarely higher than the middle of the anthers and never equal to or overtopping them	Overtopping or on the level of the anthers.	Ending at the same level as the tips of the anthers to slightly overtopping them or well overtopping them
Habitats	Short mountain turf, near melting snow	Meadows and open places in the forests	Grassy clearings between and near forests or sparse shrubs
Flowering time	III–VII	II–III	II–III(–IV)

**Table 3 plants-12-02420-t003:** Comparisons between the leaf anatomical parameters in *Crocus heuffelianus*, *C. veluchensis* and *C. tommasinianus*.

Character	*C. heuffelianus*	*C. veluchensis*	*C. tommasinianus*
Leaf width, μm	2701–4645	1673–2546	2821–3369
	3305 ± 433	2121 ± 78.6	3042 ± 245.2
Keel height, μm	562–1458	505–1157	691–979
	1024 ± 170.1	788 ± 45.6	787 ± 120.8
Keel width, μm	751–976	615–1240	553–894
	890 ± 1	817 ± 40.2	793 ± 137
White stripe, μm	836–1076	303–899	538–667
	927 ± 55.9	596 ± 48.5	591 ± 64.1
White stripe/Section ratio, %	19–36	18–34	15–23
	29 ± 4	25 ± 2.5	20 ± 2.3
White stripe/Keel ratio, %	87–138	47–115	60–98
	105 ± 8.7	78 ± 7.2	76 ± 14.2
Leaf blade thickness, μm	119–243	81–247	78–162
	176.7 ± 34.7	155 ± 28.6	119.4 ± 24.6
Palisade parenchyma thickness, μm	51–105	28–103	15–62
	76 ± 14.2	65.7 ± 13.7	31.7 ± 14.8
Pallisade cell rows	2	2	1
Palisade cells, height, μm	29–60	24–51	19–31
	43.1 ± 3.8	37.2 ± 5	24.8 ± 3.2
Spongy parenchyma thickness, μm	10–97	34–106	42–88
	58.3 ± 19.6	66.9 ± 27.4	56.9 ± 10.5
Adaxial epidermis cells length, μm	147–766	157–867	322–562
	360.3 ± 126.9	434 ± 98.9	422.8 ± 91.4
Adaxial epidermis cells width, μm	13–31	9–41.4	12.2–23
	21.5 ± 3.4	21.4 ± 3.8	17.9 ± 4.6
Adaxial epidermis cells height, μm	12–31	11–32	15–21
	23.1 ± 3.4	21.2 ± 2.7	17.3 ± 2.12
Abaxial epidermis cells length, μm	32–252	25–282	78–265
	109.4 ± 39.7	90 ± 33	142 ± 42.5
Abaxial epidermis cells width, μm	13–39	9–46	16–25
	24.3 ± 4.3	24.9 ± 4.9	20 ± 2.6
Abaxial epidermis cells height, μm	14–26	10–31	10–21
	18.2 ± 3.5	19.5 ± 3.4	15.5 ± 5.2
Stomata length, μm	20–37	20–41	21–33
	29.4 ± 3.6	30.1 ± 3.9	26.34 ± 3
Stomata width, μm	15–31	15–33	15.4–25.2
	22.8 ± 3.4	23 ± 2.6	20.1 ± 2.6
Stomata depth, μm	6–14	5–21	9–11
	9.1 ± 2	11 ± 2.2	9.2 ± 1.2
Stomata, count/mm^2^	200–375	200–400	222–400
	269.6 ± 59	295.7 ± 34.9	300.9 ± 48.5
All vascular bundles height, μm	35–158	21–268	30–150
	84.5 ± 34.2	86.3 ± 38.2	80.9 ± 35.1
Count of vascular bundles in section	19–42	20–25	21–28
	27.7 ± 13.2	22.8 ± 1.7	25.3 ± 3.1
All vascular bundles height, μm	35–177	17–153	25–114
	84.1 ± 33.9	58.3 ± 23.8	63.2 ± 24.4
All vascular bundles width, μm	29–105	17–153	25–114
	60.4 ± 18.7	58.3 ± 23.8	63.2 ± 24.4
Vascular tissue height, μm	10–111	13–168	14–92
	47.5 ± 21.7	52.2 ± 20.3	51.6 ± 23.7
Vascular tissue width, μm	19–87	17–138	21–74
	47.6 ± 17.6	53.7 ± 21.2	51.6 ± 23.7
Vascular tissue section, μm^2^	143–7585	203–17,342	230.9–4606.4
	2029.6 ± 1652.2	2582.5 ± 1901.5	2239.3 ± 1484.4
Vascular tissues related to the vascular bundle section, %	9–112	18–84	56–80
	37.1 ± 16.6	44.9 ± 10.1	63.6 ± 10.8
Terminal vascular bundles, height, μm	79–149	73–181	110–131
	116.2 ± 18.6	116.3 ± 13.2	118.5 ± 9.6
Terminal vascular bundles, width, μm	66–90	50–161	76–95
	79.3 ± 6.4	91.6 ± 10.4	83.5 ± 8.2
Terminal vascular bundles, section, μm^2^	4768–9591	203–22,319	6912–9774
	7285.3 ± 1355.6	8952.5 ± 2064.3	7802.3 ± 1325.3
Terminal vascular bundles related to the maximum size, %	13–69	36–99	56–80
	49.2 ± 11.4	70.5 ± 9.8	63.6 ± 10.8
Keel vascular bundles, height, μm	118–177	70–268	123–150
	140.6 ± 18	150.9 ± 25.8	136.5 ± 10.3
Keel vascular bundles, width, μm	70–105	47–170	93–114
	88.6 ± 10.2	105 ± 13.8	102.3 ± 8.3
Keel vascular bundles, section, μm^2^	7092–13,901	2583–32,204	3562.6–5112.9
	9862.6 ± 2182.1	13,742.4 ± 3602.2	4547.5 ± 649.7
Keel vascular bundles, related to the maximum size, %	58–100	32–100	29–42
	77.8 ± 17.1	80.3 ± 18	37.1 ± 5.3
Corner vascular bundles, height, μm	44–83	44–148	84–104
	43.6 ± 11.5	89.9 ± 14.9	91 ± 8
Corner vascular bundles, width, μm	31–66	38–109	54–72
	43.6 ± 15.7	74.7 ± 12.7	63.6 ± 7.4
Corner vascular bundles, section, μm^2^	1193–4302	1354–10,404	3562–5112
	2003.2 ± 911.4	5558 ± 1724	4547.5 ± 649.7
Corner vascular bundles, related to the maximum size, %	9–31	9–61	29–41
	15.7 ± 7	39.1 ± 8.9	37.1 ± 5.3

## Data Availability

Data is contained within the article.

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
