# Peer review of "Crocus heuffelianus—A New Species for the Bulgarian Flora from Series Verni (Iridaceae)"

_plants, 2023, doi:10.3390/plants12132420_

Round 1
Reviewer 1 Report
The article is well written and explain the research done to identify the new found apopulations. I ha done minor corrections to the text. In my opinion this MS can be accepted after minor revision

Only a minor English revision is needed
Author Response
Thank You very much for the review and the corrections in the text.
Reviewer 2 Report
This paper needs a major revision. You stated on lines 45-48 that the 1982 taxonomic revision for this group treated C. heuffelianus as a synonym of C. vernus. Figure 10 confirms that no significant differences were found to distinguish between these two taxa. With that in mind, either you should change your treatment from calling your plants C. heuffelianus to using C. vernus OR explain in detail how one can clearly distinguish between these two taxa (by means other than geographic range or habitat preferences). This will require additional morphological study, unless someone has already published a list of distinguishing traits that you can test with your existing data.
Until one of those two approaches is taken, I think it would be premature to give this manuscript a full review.
One other small point, what is C. speciosus (see line 105)? No other mention is made of this taxon in your paper.
Most sentences read fairly well, but I did notice room for further improvement.
Author Response
Thank You very much for the review. We agree with the comments.
1. We expanded the morphological section and added some more comments to the discussion. In brief, the population in Bulgaria is similar to that in Serbia. According to Racca et al. (2023), the plants in Serbia are allotetraploids, and our specimens are very similar to their description. Thank you very much for the note.
2. The name C. speciosus was written by error, done mechanically by me. Of course, we talk about C. heuffelianus. Thank you for the remark.
Reviewer 3 Report
The introduction is written in a somewhat erratic manner it is difficult to follow the authors on the taxonomic affiliations of the taxa studied... It would be better to present the taxonomic history of C. heuffelianus in more detail. Because the ITS sequences are identical in C. vernus and C. heuffelianus, it would be important to present the differences (morphological, cytological and others) between these species. The work lacks the most important feature for the C. heuffelianus - the chromosome number for the Bulgarian accessions. In general, this feature is not mentioned at all in the text.
There are small misprints in the text. E.g. in Table A1. Harpk et al. should Harpke et al.
Author Response
Thank You very much for the review.
- We compared our specimens to those described by Racca et al. (2023). Our collections seem to be similar to the tetraploids from the Western Balkans, signed as C. cf. heuffelianus. We explained our viewpoint in the discussion. Thank you for the note, it helped us.
- Thank you. We checked the manuscript for typographic errors again.
Round 2
Reviewer 2 Report
Dear Editor, I downloaded the manuscript and read the revisions including the key addition of Table 2. The revisions resulted in clear improvements. I do believe it is now at a stage where it warrants publication, with one small addition. On line 58, I would suggest that the authors insert the following line (in bold) to make it clear that more is now known than when Mathew published his work in 1982: <<However, more recent studies support the recognition of C. heuffelianus as a distinct taxon [including pertinent references].≫This short addition would give the readers a better transition to the paragraph that follows.
Author Response
Thank you very much for the kind collaboration.
We accept the suggestion.